# Genetic Counselling Needs for Reproductive Genetic Carrier Screening: A Scoping Review

**DOI:** 10.3390/jpm12101699

**Published:** 2022-10-11

**Authors:** Samantha Edwards, Nigel Laing

**Affiliations:** Harry Perkins Institute of Medical Research and Centre for Medical Research, University of Western Australia, QEII Medical Centre, Nedlands, WA 6009, Australia

**Keywords:** genetic counselling, carrier screening, reproductive health, personalized medicine, population screening, education

## Abstract

Reproductive genetic carrier screening provides individuals and couples with information regarding their risk of having a child affected by an autosomal recessive or X-linked recessive genetic condition. This information allows them the opportunity to make reproductive decisions in line with their own beliefs and values. Traditionally, carrier screening has been accessed by family members of affected individuals. In recent years, improvements to accessibility and updates to recommendations suggest that all women planning or in early pregnancy should be offered reproductive genetic carrier screening. As uptake moves towards the population scale, how can the genetic counselling needs of such large-scale screening be met? A scoping review of the literature was performed to ascertain what the genetic counselling needs of reproductive genetic carrier screening are, and what future research is needed. Four broad themes were identified in the existing literature: (1) The offer—when and in what context to offer screening; (2) Information—the importance of and what to include in education, and pre- and post-test counselling; (3) Who and how—who the genetic counselling is performed by and how; (4) Personalization—how do we find the balance between standardized and individualized approaches? Based on the existing literature, we present a set of recommendations for consideration in implementing population-scale reproductive genetic carrier screening as well as suggested areas for future research.

## 1. Introduction

Research suggests that 300 million people worldwide are affected by rare diseases [1]. Genetic conditions account for around 72% of these rare diseases and 70% of those start in childhood [1]. Nearly 90% of parents with affected children have no prior family history or knowledge of their risk [2]. The impact of these genetic conditions on families cannot be overstated.

Carrier testing has been used for many years by family members of individuals affected by a recessive genetic condition such as cystic fibrosis (CF), spinal muscular atrophy (SMA), or fragile X syndrome (FRAX) to determine their carrier status [3,4,5]. In this “cascade screening”, family members would be tested for the known familial biochemical change [6] or variant, to identify whether or not they are carriers of the family recessive disease and at risk of having a similarly affected child. This mode of carrier testing necessitates that at least one family member has experienced what can be a diagnostic odyssey for many years prior to identifying the clinical diagnosis and familial variant [7]. Those undergoing cascade screening will generally have some lived experience of the genetic condition for which they are using carrier screening. 

Population-based carrier screening was first performed for genetic conditions with high carrier frequencies, particularly in ethnic populations, such as sickle cell anaemia or thalassaemia in Mediterranean populations, or Tay Sachs disease in the Ashkenazi Jewish population [6]. Single disease, single gene, or small multi-disease genetic variant or gene screening panels have been developed and made available to relevant individuals in these populations [8,9]. A high proportion of these individuals too would have some experience of or exposure to the genetic conditions they were being screened for. However, it is now recognized that reliance on ethnicity or family history alone to determine the need for carrier screening leads to the under-identification of at-risk carrier individuals and couples [10].

It has been possible in Australia to access a basic reproductive genetic carrier screening (RGCS) panel inclusive of CF, SMA and FRAX since 2012 on a user-pays basis [2]. However, the limited panel leaves individuals and couples unaware of their chance of having a child affected with one of the thousands of other recessive genetic conditions they could be at risk of passing on. 

More recently, the advances in genomic sequencing technology and its increasing affordability have made it possible to offer the large panel screening of 100 s to 1000 s of genes at once [6,11]. These RGCS panels are mostly offered on a user-pay basis and can provide higher levels of reassurance for couples regarding their risk of having a child affected by a genetic condition [6]. They also allow the opportunity for reproductive decision-making in line with an individual or couple’s beliefs and values [12]. However, the sheer number of conditions included, and the largely healthy population accessing screening with no lived experience of the conditions for which they are being screened, increases by orders of magnitude the complexity of the required genetic counselling [13].

Over recent years, professional bodies in a number of countries have released recommendations advising that all women should be offered genetic carrier screening when planning a pregnancy or in early pregnancy [12,14,15,16,17,18]. Many of these recommendations also recognize that the complexities of RGCS necessitate comprehensive genetic counselling to support individuals and couples in their decision-making both pre- and post-test. 

In 2021 the American College of Medical Genetics (ACMG) [13] released practice guidelines relating to RGCS and highlighted that traditional genetic counselling methods can be longwinded and likely lack feasibility at the population scale. Despite the ever-increasing number of accredited genetic counsellor training courses and graduates entering the workforce, there are simply not enough genetic counsellors to meet the demand for pre- and post-test counselling for RGCS at the population scale [13,19]. 

The increasing push to offer RGCS programs at the population scale raises questions relating to how the substantial genetic counselling requirements for such programs can be adequately met. What are the genetic counselling requirements specifically? Who should be involved in providing genetic counselling for RGCS programs and at which stages in the process? Are there alternative methods for performing genetic counselling? Where should the specialist skills and experience of genetic counsellors be focussed within the scope of RGCS service delivery at scale? To answer these questions the genetic counselling requirements of genetic carrier screening must be explored and understood. 

## 2. Materials and Methods

This scoping review was conducted based on the Arksey and O’Malley [20] framework and included five key phases: (1) identifying the research question, (2) identifying relevant studies, (3) study selection, (4) charting the data, and (5) collating, summarizing and reporting the results. 

This review was guided by the question, ‘What are the genetic counselling requirements for RGCS?’ where ‘RGCS’ refers to expanded gene panels and whole exome sequencing with targeted computational analysis [6] for autosomal recessive and/or X-linked genetic conditions. For the purpose of this study, a scoping review is defined as a type of research synthesis that aims to ‘map the literature on a particular topic or research area and provide an opportunity to identify key concepts, gaps in the research, and types and sources of evidence to inform practice, policymaking and research’ [21].

A search was conducted in three electronic databases; MEDLINE/PubMed, EMBASE and CINAHL/EBSCO. The databases were selected to be comprehensive and to cover the relevant disciplines. No limits on date, language, or type were placed on the database search. The search query consisted of terms relating to three key areas: genetic counselling, carrier screening and reproductive health/behaviour. The search query tailored these three subject areas to the specific requirements of each database (Appendix A). Unpublished articles or grey literature were excluded from the search parameters.

A two-stage screening process was used to assess the relevance of studies identified in the search. All citations were imported into the web-based bibliographic manager Mendeley Reference Manager for the subsequent title and abstract relevance screening. The first level of screening involved a review of only the title and abstract of citations to identify those studies that met the minimum inclusion criteria. Exclusion criteria included duplicate articles, unpublished articles or grey literature, ineligible article type (e.g., review, commentary, editorial, committee opinion, practice guidelines, etc.), references to screening unrelated to RGCS (e.g., predictive testing, presymptomatic/predisposition testing, aneuploidy, noninvasive prenatal testing (NIPT), etc.), or where there were no genetic counselling outcomes. 

The second level of screening involved a full review of the article for exclusion or inclusion in the data characterisation phase. All citations deemed relevant after the title and abstract screening were obtained for subsequent review of the full-text article. Study characteristics such as publication year, research characteristics, participant type, and carrier screening type were extracted into a single spreadsheet for descriptive statistical analyses. 

The researcher adopted a narrative synthesis approach due to methodological heterogeneity across the reviewed studies. This allowed for the organisation, description, exploration and interpretation of the study findings relating to genetic counselling outcomes [22], each of which was identified and coded. Inductive thematic analysis was applied to the coded data followed by idea mapping [22]. 

The database search, article review, data characterisation and evidence synthesis were conducted by a single researcher. The scoping review was registered on Open Science Framework (https://doi.org/10.17605/OSF.IO/MR8ZK, accessed on 21 June 2022) along with the protocol (https://doi.org/10.17605/OSF.IO/72YKR, accessed on 3 October 2022). Specific ethics approval for this scoping review was not required.

## 3. Results

As outlined in Figure 1, the initial search conducted in May 2022 yielded 880 potentially relevant citations. Following deduplication and relevance screening, 72 citations met the inclusion criteria based on title and abstract, and the corresponding full-text articles were procured for review. After a review of the full-text articles, only 37 remained for data characterisation and were included in the analysis (Appendix A). 

Data characterisation (Table 1) identified articles published from 1994 to 2022 with 78.4% published in the last 7 years. Studies originated in multiple regions including Australia, Taiwan, and various European countries, but 54.1% of the studies originated in the United States. Quantitative research methodologies were most prominent (62.1%) as was reliance on survey data (54.1%), however, qualitative interview and focus group data accounted for 32.4% of the studies. The research was primarily retrospective (78.4%) with a range of stakeholder participant groups including prospective users (16.2%), retrospective users (62.2%), genetic healthcare professionals (HCPs) (16.2%) and non-genetic HCPs (8.1%). Of the 37 studies, 81.1% were related to RGCS with an expanded panel. The remaining seven articles were based on single-gene (e.g., CF, SMA) or small panel (haemoglobinopathies) population screening programs but were included for the relevance of genetic counselling outcomes. 

Inductive thematic analysis and idea mapping [22] identified four broad themes (Appendix A): The Offer—when and in what context to offer screening;Information—the importance of and what to include in education, and pre- and post-test counselling;Who and how—who the genetic counselling is performed by and how;Personalisation—how do we find the balance between standardized and individualized approaches?

### 3.1. The Offer

A number of genetic counselling outcomes identified pertained to when and under what context RGCS should be offered. Of the reviewed articles, 14 identified preconception over early pregnancy as the ideal time to offer RGCS [24,25,26,27,28,29,30,31,32,33,34,35,36,37]. Amongst these 14 studies, all four participant groups were in favour of preconception carrier screening. These included 48% of the retrospective user studies, a third of genetic and non-genetic HCP studies and 17% of prospective user studies.

Two studies highlighted the benefits of offering RGCS to couples [24,27] and two studies specifically pointed out challenges associated with offering individuals RGCS [27,34]. The benefits of couple screening and issues with screening individuals were identified by studies involving retrospective users and genetic HCPs. None of the reviewed articles recommended offering RGCS exclusively to couples or individuals. Rather, there was a general recognition that offering RGCS in both contexts was advantageous and appropriate.

### 3.2. Information

The necessity for information surrounding RGCS was a common thread through many of the genetic counselling outcomes identified in the reviewed articles. Education for the general public was said to be warranted by 24% of reviewed articles. Interestingly, this outcome was restricted to RGCS user studies, including three prospective user and six retrospective user studies. In contrast, 41% of all reviewed articles highlighted the importance of RGCS education and training programs for HCPs, including around a third of prospective [32,38] and retrospective user studies [24,28,31,34,39,40,41], over 80% of genetic HCP studies [30,42,43,44,45] and 67% of non-genetic HCP studies [31,32,33,34,35,36,37,38,39,40,41,42,43,44,45,46].

Forty-three percent of reviewed articles agreed that pre-test counselling for RGCS should be thorough and consistent, including two prospective user [32,47], 11 retrospective user [29,36,37,39,41,48,49,50,51,52,53] and three genetic HCP [30,43,44] studies. Forty-three percent of articles agreed that post-test counselling for RGCS should be thorough and consistent, including one prospective user [54], 11 retrospective user [26,28,29,33,36,37,39,40,49,52,55] and four genetic HCP [30,43,44,45] studies. It is interesting to note that none of the studies involving non-genetic HCPs emphasised thorough and consistent genetic counselling for RGCS as important.

Information of particular importance to include in pre-test counselling was also identified. Five of the reviewed articles, including prospective user [54], retrospective user [33,50] and genetic HCP [42,45] studies, emphasised personal health implications to carriers as an important inclusion for pre-test counselling of RGCS. Seven studies involving retrospective users [34,56], genetic HCPs [30,45,57,58] and non-genetic HCPs [46,58] agreed the clinical significance and utility of results should be covered in pre-test counselling also.

A number of considerations were deemed important for inclusion in post-test counselling. Three genetic HCP studies emphasized the need for RGCS results to be carefully, consistently and thoroughly researched prior to result delivery [27,30,45]. Accessibility to specialist clinicians [48], support groups and patient families [27] were also recommended by two of the reviewed articles. Sensitivity and exploration around the reproductive options available to carrier couples and individuals was raised by 35% of all reviewed articles, with all stakeholder groups represented [25,28,29,30,33,36,48,49,53,54,55,57,58].

The importance of including the limitations of RGCS were raised by 32% of the reviewed articles including one prospective user [54], 9 retrospective user [26,34,36,37,41,50,53,55,56] and two genetic HCP [27,57] studies. A single genetic HCP study [57] and four retrospective user studies [26,36,37,50] recommended that residual risk should be articulated during both pre- and post-test counselling. Surprisingly only two studies emphasised the need to distinguish between RGCS and other screening tests offered during pregnancy, such as NIPT and ultrasound scans [34,57].

### 3.3. The Who and How

Many of the genetic counselling outcomes related to who was performing the pre- and post-test counselling for RGCS. Whilst some studies specifically utilized genetic HCPs in the offer of RGCS [24,29,30,40,44,45,48,51,58], some involved non-genetic HCPs such as obstetricians, gynaecologists and general practitioners [31,36,43,46,50,52]. There were three studies that identified HCPs lacking in knowledge and confidence regarding RGCS as an area for concern [43,45,46]. Nineteen per cent of the reviewed articles, all of which were retrospective user studies, found that the attitude of the HCP offering RGCS can influence patient decision-making in relation to the uptake of screening and subsequent reproductive decisions [26,34,35,36,41,48,53]. Almost 40% of the studies, including eight retrospective user [26,28,31,34,39,40,48,49], 5 genetic HCP [27,30,43,45,57] and two non-genetic HCP [31,46] studies referred to the importance or underutilization of referral to genetic HCPs for at-risk or carrier couples or individuals following RGCS.

Following the emphasis on education in the previous ‘Information’ theme, 30% of all studies reviewed mentioned the involvement of genetic HCPs in the development of education and support resources/tools for RGCS, specifically one prospective user study [32], 6 retrospective user studies [28,29,34,35,39,40] and unsurprisingly four genetic HCP studies [27,30,43,45].

A number of studies recognized the utility of alternatives to traditional face-to-face genetic counselling. Alternative methods such as written information leaflets, videos, and telehealth were used or mentioned by 46% of studies involving all stakeholder groups for the purposes of pre-test counselling [24,26,27,28,31,32,33,36,37,38,39,40,44,49,53,54,59]. In contrast, only 16% of studies mentioned or used similar alternative methods in relation to post-test counselling for RGCS [24,25,30,31,39,40] with all stakeholders represented, bar the prospective users.

### 3.4. Personalisation

Despite the apparent emphasis on thorough and consistent information during pre-test counselling for RGCS, 46% of the reviewed articles, including all stakeholder groups, also recognized the need for some degree of individualisation of the process and the delivery of genetic counselling [25,32,34,35,36,38,39,41,44,47,49,51,52,53,54,57,58]. Twenty-two per cent of the reviewed articles, including all stakeholders, emphasised the importance of considering the patient’s beliefs and values when counselling around RGCS [35,36,38,41,48,53,55,58]. The context under which a patient might be considering RGCS should also be taken into account and will influence the genetic counselling process [27,34]. The necessity for making a tailored result delivery plan with each patient was identified by three retrospective user [25,36,49], 1 non-genetic HCP [58] and two genetic HCP [45,58] studies. All stakeholder groups agreed that pre- and post-test counselling for RGCS should include and be sensitive to the accessibility of reproductive options available to the patient, with 35% of reviewed articles raising this as a genetic counselling outcome [25,28,29,30,33,36,48,49,53,54,55,57,58].

RCGS user group studies found that the lived experience of the patient influenced and potentially complicated their comprehension of the screening test result [25,34,35,36,51,53,60]. A single retrospective user study emphasized the relief, reassurance and confidence that patients may feel upon receiving a negative or low-risk RGCS result [50].

## 4. Discussion

The reviewed studies indicated that the offer of RGCS is largely concerned with the concepts of screening the individual [27,34] vs. the couple [24,27], and screening in preconception or early pregnancy [24,25,26,27,28,29,30,31,32,33,34,35,36,37]. It was recognized that offering RGCS to couples streamlines the genetic counselling process. When RGCS is offered sequentially, it was noted that engaging both partners at the pre-test counselling stage facilitates expedient partner screening when necessitated [27]. Carrier screening offered concurrently, and/or reporting on a couple basis, reduced the demand for genetic counselling as only at-risk couples require thorough post-test genetic counselling [24,27]. As a result, it may be most prudent to offer RGCS at the population scale predominantly on a couples basis due to the limited genetic counsellor resources available.

There are scenarios where screening of individuals is necessary, for example, in the case of anonymous donors and where a reproductive partner is unknown or unavailable. However, screening of individuals was considered problematic in some studies as genetic counselling for individual carrier status places a huge burden on genetic counselling resources and will not always be relevant to reproductive decision-making [27]. As evidenced by Bell et al. [61], where a screening panel of 437 genes identified 70% of participants as carriers, the sheer volume of genetic counselling required for individual carrier status as a result of RGCS at scale will outstrip the genetic counsellor resources available [13,19,30]. Similarly challenging is when genetic carriers are identified via sequential screening offered in early pregnancy but the partner is not available for screening or refuses screening. Such scenarios result in complex genetic counselling regarding reproductive options available and may potentially lead to unnecessary prenatal testing [27,34]. The possibility of unnecessary prenatal testing should be included in pre-test genetic counselling.

Offering RGCS preconception was recognized as preferable by all stakeholders. Preconception RGCS allows at-risk carriers adequate time for appropriate genetic counselling, to adjust to their result, consider the reproductive options available to them and factor in their own personal beliefs and values [24,25,26,27,28,29,30,31,32,33,34,35,36,37]. To make RGCS available at scale, considerable effort should be made to increase the likelihood of RGCS being accessed preconception, to minimize the impact on genetic counselling resources. At the same time, the studies reviewed recognized that it is also important and often more convenient to offer RGCS during early pregnancy. However, RGCS in early pregnancy potentially leads to acute, time-constrained and complex post-test counselling for at-risk carriers, which genetic counsellors may be best placed to manage [25,27,28,29,30,31,34,35,36,37].

The majority of reviewed studies agreed that thorough and consistent RGCS information sources should be available at all time points, including general education, and at pre- and post-test counselling. In some regions, it is in fact mandated that pre-test counselling should be performed by a qualified medical specialist before the ordering of genetic tests [62]. Broadly, it was accepted that further RGCS education of both the public [24,32,34,35,38,41,48,49,60] and non-genetic healthcare professionals [24,28,30,31,32,34,38,39,40,41,42,43,44,45,46] was needed. This is similar to research surrounding NIPT and critical information not being received by patients when counselling is performed by non-genetic healthcare professionals—which highlights the importance of thorough education for non-genetic healthcare professionals [63]. In some studies, education, even for genetic healthcare professionals, was identified as important, particularly when reproductive genetics was not their main field, or where rare and complex carrier results necessitated careful and thorough research prior to post-test counselling [30,42,44,45].

Many of the reviewed studies identified and agreed upon information that should be included in pre-test counselling for RGCS. A point of difference between single gene screening and expanded carrier screening panels is the level of specificity one can employ when providing pre-test counselling. Whilst it is relatively simple to give a detailed explanation of a single gene or genetic condition, it is much more difficult, in reality, impossible to do this at the same level when a panel includes hundreds of genes. In addition to being a time-consuming and complex undertaking, a detailed description of hundreds of genes has the potential for information overload during the consent process, which has led to a preference for generalised information during pre-test counselling [12]. Aside from an explanation of RGCS and a broad description of the genetic conditions screened for, particular issues identified for inclusion were: the limitations of RGCS [26,27,34,36,37,41,50,53,54,55,56,57], how it differs from other antenatal screening [34,57], residual risk [26,36,37,50,57], the potential for results with uncertain clinical utility [30,34,45,46,56,57,58], the possibility of personal health implications for carriers [33,42,45,50,54], and the existence of reproductive options [25,28,29,30,33,36,48,49,53,54,55,57,58].

Information that should be included in post-test genetic counselling was also identified. It has been noted that a generalised approach to information during pre-test counselling can lead to the misunderstanding of RGCS results [12,57,64,65]. Thus, it was recommended that the limitations of RGCS [26,27,34,36,37,41,50,53,54,55,56,57] and residual risk [26,36,37,50,57] be reiterated in post-test counselling for all screening results. As emphasized by Dive and Newson, it is of particular importance that all patients undertaking RGCS understand that a ‘negative’ result does not guarantee future healthy babies [26,36,37,50,57,66]. Similarly, it is imperative that patients understand that carrier screening does not negate the need for other screening options offered in pregnancy [34,57], such as NIPT and ultrasound. For carriers and at-risk couples, reproductive options [25,28,29,30,33,36,48,49,53,54,55,57,58] should be discussed, and where relevant, the clinical utility of their result [30,34,45,46,56,57,58], and any personal health implications [33,42,45,50,54]. Some studies also identified the importance of at-risk couples accessing clinician [27,48], support group and affected family [27] advice relevant to the condition for which they are genetic carriers.

This review of the literature identified an overwhelming desire for thorough standardized RGCS information to be available for public and healthcare professional education [24,28,30,31,32,34,35,38,39,40,41,42,43,44,45,46,48,49,60] and throughout the screening process [24,26,28,29,30,31,32,33,34,35,36,37,38,39,40,41,42,43,44,45,46,47,48,49,50,51,52,53,54,55,60]. Genetic healthcare professionals, including genetic counsellors, would be integral to developing resources and education programs that meet these needs at the population scale. Given that genetic counsellors are recognized as a limited resource, who is carrying out the pre- and post-test counselling for RGCS at the population scale is an area of concern.

The studies reviewed broadly accepted that pre-test genetic counselling can be adequately performed by non-genetic healthcare professionals such as primary care physicians, obstetricians and gynaecologists. Non-genetic HCPs may also be able to perform post-test genetic counselling for non-carrier results and the more common at-risk couple results. However, it was also recognized that a lack of confidence, or knowledge, will influence the adequacy of their involvement [43,45,46], as similarly stated by Liehr in relation to the offer of NIPT [63]. An emphasis on RGCS education for the public will likely reduce the time required to perform pre-test counselling, making it easier for non-genetic HCPs to offer RGCS during their time-constrained consults. Similarly, RGCS education and training for HCPs will increase both their knowledge and confidence in offering RGCS and reduce the burden on genetic HCPs. This in turn would allow the finite genetic HCP resources to be concentrated on referrals for complex carrier results for at-risk individuals and couples [26,27,28,30,31,34,39,40,43,45,46,48,49,57].

A number of studies successfully utilized supporting resources, such as videos and information leaflets, as part of the pre-test counselling process [24,26,27,28,32,33,36,37,38,39,40,44,48,49,53,54,59]. In some cases, result delivery and post-test counselling were performed by letter [36,43]. There was no evidence of interactive methods such as decision aids, chatbots or online learning being employed. There was also no evidence of programs with accessible genetic counsellor support provided alongside the non-genetic healthcare offer of RGCS. Pre-test counselling is often performed in time-poor consultations by non-genetic healthcare professionals [28,31]. The integration of alternative communication methods may address the genetic counselling needs of large-scale RGCS programs. This could streamline the information-giving process through the utilisation of supportive resources and facilitate the escalation of queries and concerns to genetic counsellors.

Healthcare professional attitudes were identified as influential in decision-making and highlighted the need for neutral language and non-directiveness during RGCS genetic counselling [26,34,35,36,41,48,53]. It is widely accepted that genetic counsellors are specifically trained to use non-directive counselling techniques and neutral language [67] to enhance patient autonomy [68]. Adopting this form of communication may be difficult for non-genetic healthcare professionals, but it is important if individuals and couples being offered RGCS are to make informed decisions in line with their own beliefs and values. In the absence of non-directive counselling, the offer of RGCS can become prescriptive, and couples may experience unnecessary regret when healthcare professional attitudes have influenced reproductive decisions [26,34,35,36,41,48,53]. Thus, an emphasis on patient autonomy through non-directive communication and the use of neutral language should be included in education relating to RGCS.

Whilst many of the studies agreed that RGCS information should be standardized for pre-test counselling and at least have a standardized basis for post-test counselling, there was a consistent call for personalisation in the genetic counselling process. Individual and couple attitudes towards, and decision-making around RGCS, will be influenced by factors specific to the individual and couple. The individual or couple context will influence how RGCS is offered, and the beliefs and values of patients will impact their decision-making throughout the RGCS process [35,36,38,41,48,53,55,58]. Patient accessibility and preferences should be considered when planning result delivery and when discussing reproductive options [25,36,45,49,58]. The lived experience of an individual or couple will affect their ability to comprehend and adjust to their RGCS result [25,34,35,36,51,53,60]. An individualised, or tailored approach to pre-test counselling, result delivery and post-test counselling, may be difficult to achieve when RGCS is being offered by non-genetic healthcare professionals. How can enhanced patient autonomy be achieved [68] at scale during time-poor non-genetic HCP consultations and with limited genetic counsellor resources? How can an acceptable balance be achieved? Increased public education around RGCS and the use of standardized and interactive support materials during pre-test counselling could allow time-poor non-genetic HCPs to focus on personalizing the offer of RGCS to each patient’s circumstances. The integration of genetic counsellor support for HCPs and patients undergoing RGCS and the ability to escalate to referrals could also allow for personalization throughout the pre-test counselling process.

The reviewed literature overwhelmingly agreed that RGCS should ideally be offered by non-genetic healthcare professionals to couples in preconception, where the results would inform future reproductive decisions. Sufficient and appropriate pre- and post-test genetic counselling is vital in providing enhanced patient autonomy and should be accessible through suitably trained healthcare professionals. However, it must be recognized that many non-genetic healthcare professionals are offering RGCS during time-poor consultations that are not conducive to a personalized approach. Currently, RGCS is largely available through commercial providers [69,70] with a business model potentially at odds with the population’s need for information and knowledge [71]. Who then is responsible for ensuring the quality of genetic counselling and the information that patients receive when they are offered RGCS? How can it be ensured that standardized information is available and accessible to all patients, yet they also receive personalized support throughout their decision-making process? How best can the finite resources of genetic counsellors and genetic specialists be used when RGCS is offered at scale?

## 5. Recommendations

The genetic counselling requirements for RGCS are complex and implementing them at the population scale is challenging, but it is an issue that must be considered sooner rather than later. Based on the reviewed literature, we propose the following considerations when implementing RGCS programs at scale to ensure: (1) adequate and consistent genetic counselling is provided, (2) the burden on genetic counselling resources is minimized, and (3) genetic counsellor skills are utilized for maximum effect:An offer of RGCS should include sufficient and appropriate pre- and post-test genetic counselling.RGCS should be offered preconception as much as possible in order to facilitate streamlined genetic counselling and minimize the need for acute and complex genetic counselling.RGCS should also be offered in early pregnancy accompanied by appropriate genetic counselling.Screening at scale should be offered on a couple basis, as it reduces the demand for genetic counselling resources and specifically provides information useful to reproductive decision-making. However, there will be contexts where RGCS of individuals is required and this should be made available.Genetic healthcare professionals, including genetic counsellors, should be involved in the development of standardized RGCS education resources for both the public and healthcare professionals.Genetic healthcare professionals, including genetic counsellors, should be involved in the development of consistent and thorough RGCS resources to be used as standardized supportive materials during pre- and post-test counselling.RGCS should be offered by primary care physicians, obstetricians, gynaecologists and other relevant non-genetic healthcare professionals.Supporting resources and the ability to escalate queries to a genetic counsellor should be integrated into the RGCS pre- and post-test counselling processes.Carrier individuals and carrier couples with complex genetic counselling needs should be identified and immediately referred to a genetic counsellor.Carrier individuals and carrier couples should be given the opportunity to seek advice from clinicians and support groups with relevance to the genetic condition from which they are at risk of having an affected child.

## 6. Limitations

This scoping review was conducted by a single researcher and may thus be limited by a subjective interpretation of the data. The nature of a scoping review means that the available literature was not screened to exhaustion as a systematic review would have been. Thus, it is possible that potentially valuable studies may have been omitted by chance. Finally, the reviewed studies were conducted within varying politico-legal and cultural contexts, the specifics of which were not stated and thus assumptions may have influenced the formation of the recommendations. Implementing generalised recommendations into each specific regional context may be challenging but would offer a starting point on a potentially global journey to offering RGCS at scale.

## 7. Future Research

The review of the literature identified a distinct dearth of insight into pathology provider and policymaker views on genetic counselling requirements for RGCS. Pathology providers represent an important stakeholder group given that they perform the screening, variant curation and reporting of results. How does genetic counselling impact their processes? What are the roles of laboratory-based genetic counsellors? Similarly, standardized practices might be mandated by government regulatory bodies as they have been in Germany with the German Genetic Diagnostics Act [62], yet the literature shows little evidence of research into the perceived value of genetic counselling for RGCS to the vast majority of government bodies. The politico-legal and cultural contexts of different regions will be highly influential on the possibility for RGCS to be offered at scale [70]. Further investigation into the perspectives of these stakeholder groups is needed for the successful implementation of RGCS at scale.

Whilst it is recognized that RGCS offers based on positive family history result in the under-identification of carrier individuals and at-risk couples [12], family history itself should not be ignored. Family history, for example, may influence which RGCS panel is offered, as not all panels are created equally. Offering a panel that does not adequately cover a familial or population variant may lead to a false-negative carrier status and false reassurance. Additional questions are: How does family history factor into variant curation and reporting for pathology providers of RGCS? How is family history being gathered and reported during pre-test counselling? Specific research into the impact of family history on RGCS is needed in order to develop recommendations for minimum standards of practice during RGCS genetic counselling.

Further research is also needed into standardising and streamlining RGCS education for both the public and healthcare professionals. Evidence would suggest that healthcare professionals access RGCS education from a number of different sources, including local and regional conferences, commercial provider representatives, journal articles and colleagues. The provision of standardized RGCS education for the public would likely reduce the time burden on genetic counselling resources, as individuals would enter the pre-test counselling process with a greater knowledge base [72]. Similarly, standardized RGCS education for healthcare professionals would facilitate the provision of adequate and consistent pre- and post-test counselling. This would further minimize the burden on genetic counselling resources, leaving genetic counsellors able to focus their expert skills on complex RGCS cases requiring their specific skill set.

This scoping review of the literature identified the need for consistent standardized information during pre- and post-test counselling for RGCS and also highlighted the importance of personalized genetic counselling when necessary. Finding the right balance between standardisation and personalisation will be extremely important when implementing RGCS programs at scale. Further research into how this balance might be achieved is needed. One possibility may be integrating genetic counsellor support for both healthcare professionals and individuals/couples throughout the RGCS process, enabling the genetic counsellors to provide personalisation within a standardized system. Another area for investigation is interactive pre- and post-test counselling methods that provide an alternative to, and minimize the need for, traditional genetic counselling, e.g., decision aids, chatbots, online education, etc.

## 8. Conclusions

The genetic counselling requirements for RGCS are not novel but are significantly more complex when implemented at scale. The genetic counselling requirements of smaller RGCS programs are evident and can provide insight. Yet the specific complexities of RGCS at scale appear under-researched and should be explored further in future studies. With the push for RGCS to be offered at scale and the potential conflict between commercial and patient agendas, it is imperative for a consensus of recommendations for the implementation of RGCS programs at scale from the wider genetics community in each country. The researchers put forward the recommendations set out in this scoping review as a starting point for developing consensus.

## Figures and Tables

**Figure 1 jpm-12-01699-f001:**
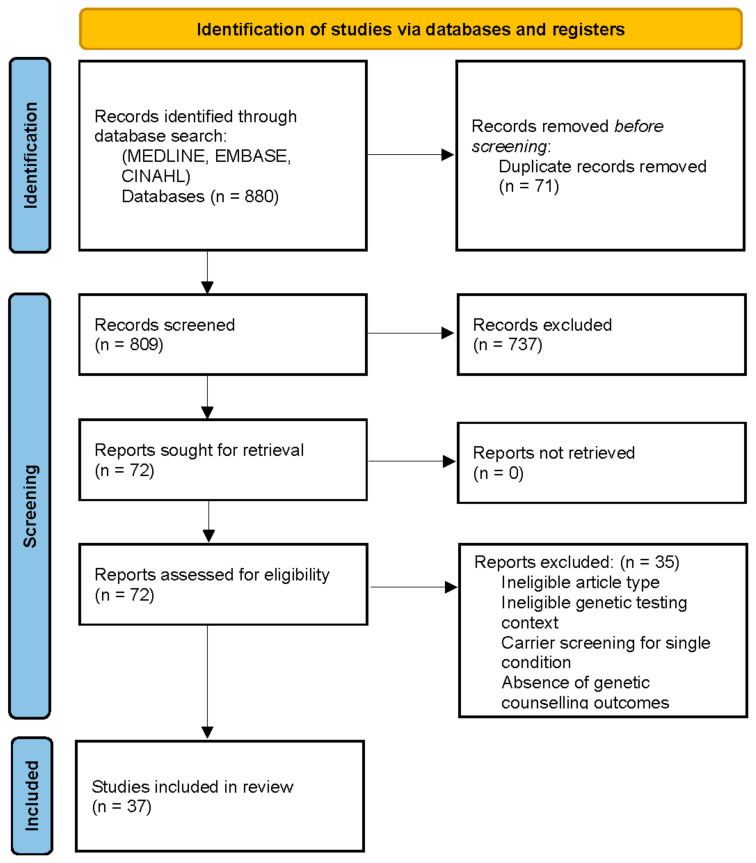
PRISMA flow diagram for the scoping review of genetic counselling needs of reproductive genetic carrier screening. Adapted from Page et al PRISMA 2020 Statement [23].

**Table 1 jpm-12-01699-t001:** General characteristics of included studies (*n* = 37).

Characteristic	Number (*n* = 37)	Percentage (%)
Publication year		
<2000	1	2.7%
2000–2009	2	5.4%
2010–2014	5	13.5%
2015–2019	19	51.4%
2020–2022	10	27.0%
Region of origin		
Australia	4	10.8%
Belgium	3	8.1%
Canada	1	2.7%
Germany	1	2.7%
Netherlands	6	16.2%
Taiwan	1	2.7%
United Kingdom	1	2.7%
USA (incl. Mexico)	20	54.1%
Research methodology		
Mixed methods	4	10.8%
Qualitative	10	27.0%
Quantitative	23	62.1%
Data types *		
Case study	1	2.7%
Chart review	7	18.9%
Focus group	3	8.1%
Interview	9	24.3%
Survey	20	54.1%
Other	1	2.7%
Research timing		
Prospective	8	21.6%
Retrospective	29	78.4%
Participant characteristics *		
Genetic healthcare professionals	6	16.2%
Non-genetic healthcare professionals	3	8.1%
Potential/prospective users	6	16.2%
Retrospective users	23	62.2%
Screening type		
Cystic fibrosis	5	13.5%
Haemoglobinopathies	1	2.7%
Spinal muscular atrophy	1	2.7%
Expanded panel	30	81.1%

* where *n* ≥ 37 as some studies included multiple data types and/or participant groups.

## Data Availability

The data presented in this study are openly available in Open Science Framework at https://doi.org/10.17605/OSF.IO/MR8ZK, accessed on 7 October 2022, under *Resources*.

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
