# Peer review of "Genetic Counselling Needs for Reproductive Genetic Carrier Screening: A Scoping Review"

_jpm, 2022, doi:10.3390/jpm12101699_

Round 1
Reviewer 1 Report
Authors did a review of literature data to answer the question how the now possible massive and potentially comprehensive reproductive genetic carrier screening (RGCS) can be comprehensively covered and accompanied by genetic counselling.
Comments:
- - Line 80 - here the abbreviation RGCS needs to be used
- - Lines 155ff – for this part authors need also to consider the legal regulations of different countries – e.g. in Europe genetic testing is regulated by ‘gene-diagnostic laws’ - this reviewer knows specifically for German law, which clearly states that before each genetic test including RGCS, a genetic counselling must be offered to the individual or couple; this is crucial, as just in this pre-genetic-test-counselling non-specialists get a chance to understand what they are testing, what they can expect and only then they can decide if they really want this test.
- - Line 161 and 1st paragraph of section ‘the who’ – here also PMID: 34220958 may be considered, which also argues that specialists need more training
- - Line 216ff – 1st senstece of #5 – this statement needs to be checked, as in Europe/ Germany the need for genetic counselling is clearly defined in German laws and as far as the reviewer knows also in other European countries. It is for sure different in e.g. USA, China, and India – thus here more differentiation is necessary and also discussion of cultural background and general views on what individual human (also unborn) life is worth in different cultures.
- - Line 250ff: in the recommendations it should be included that genetic counselling before RGCS being offered is urgently necessary and should always be done – optimally in an independent appointment with enough time for the individual or couple to think about information they got in this counselling – later couples should get a separate date for taking blood and for ordering the test.
- - For role of counselling as seen by Europeans see also Pubmed https://pubmed.ncbi.nlm.nih.gov/?term=skirton+h+counselling&sort=date
Author Response
2.1 Line 80 - here the abbreviation RGCS needs to be used
Lines 79 and 80 have been amended to include the abbreviation RGCS.
2.2 Lines 155ff – for this part authors need also to consider the legal regulations of different countries – e.g. in Europe genetic testing is regulated by ‘gene-diagnostic laws’ - this reviewer knows specifically for German law, which clearly states that before each genetic test including RGCS, a genetic counselling must be offered to the individual or couple; this is crucial, as just in this pre-genetic-test-counselling non-specialists get a chance to understand what they are testing, what they can expect and only then they can decide if they really want this test.
A sentence acknowledging that some regions have genetic testing mandates requiring pre-test counselling to be performed prior to ordering a genetic test has been added at line 235 and The German Reference Centre for Ethics in the Life Sciences (DRZE) website referenced accordingly.
2.3 Line 161 and 1st paragraph of section ‘the who’ – here also PMID: 34220958 may be considered, which also argues that specialists need more training
A sentence has been added at line 238 referring to the similar findings around the offer of NIPT by non-genetic healthcare professionals and the importance of thorough education and referencing the Liehr paper.
Liehr has been referenced again at line 275.
2.4 Line 216ff – 1st senstece of #5 – this statement needs to be checked, as in Europe/ Germany the need for genetic counselling is clearly defined in German laws and as far as the reviewer knows also in other European countries. It is for sure different in e.g. USA, China, and India – thus here more differentiation is necessary and also discussion of cultural background and general views on what individual human (also unborn) life is worth in different cultures.
This paragraph has been revised at line 366 to include reference to the German mandates on genetic testing and at line 368 to reference the influence that politico-legal and cultural contexts of a given region can have on offering RGCS at scale.
2.5 Line 250ff: in the recommendations it should be included that genetic counselling before RGCS being offered is urgently necessary and should always be done – optimally in an independent appointment with enough time for the individual or couple to think about information they got in this counselling – later couples should get a separate date for taking blood and for ordering the test.
The recommendations have been revised and the first recommendation (at line 331) now reads: An offer of RGCS should include sufficient and appropriate pre-and post-test genetic counselling. The authors agree that it is ideal for an offer of RGCS and the time of ordering the test to occur in separate appointments allowing for adequate decision-making and minimising the possibility of decisional regret or coercion as set out in Ong, et al (PMID: 31209093). However, realistically given the time-poor nature of many general physician and specialist consultations, the existence of waitlists for ‘routine’ appointments and the burdensome nature of having to attend two appointments it is unlikely that this approach will work at scale. As such we have not included this stipulation in the recommendations.
2.6 For role of counselling as seen by Europeans see also Pubmed https://pubmed.ncbi.nlm.nih.gov/?term=skirton+h+counselling&sort=date
The paragraph at line 298 has been revised to include mention of enhanced patient autonomy and references Godino et al.
Reviewer 2 Report
Thanks for having me as a reviewer of this manuscript.
It was easy to read and fairly good to understand. The material and methods are described well, and so are the results.
I am concerned about the discussion. The discussion is mainly focussing on the four broad themes, but it does not discuss them jointly, although they are certainly highly related to each other. There is commercial interest on one side, but there is also need to have enough information for the patients.
As a geneticist I know about the advantages. However, the ethical component of receiving "negative" results should be more addressed as well.
Another point is the legal situation. How are the regulations of genetics tests? Conservative governments might restrict testing, although subgroups of their population might require testing.
Author Response
Thanks for having me as a reviewer of this manuscript.
It was easy to read and fairly good to understand. The material and methods are described well, and so are the results.
3.1 I am concerned about the discussion. The discussion is mainly focussing on the four broad themes, but it does not discuss them jointly, although they are certainly highly related to each other.
A paragraph has been added at line 313 tying the four themes together.
3.2 There is commercial interest on one side, but there is also need to have enough information for the patients.
The same paragraph (line 313) identifies the potentially competing interest of the commercial providers’ business model and public need for knowledge. The revision highlights the need for someone to take responsibility for ensuring that adequate and appropriate information and genetic counselling is available to all accessing RGCS.
3.3 As a geneticist I know about the advantages. However, the ethical component of receiving “negative” results should be more addressed as well.
Line 257 and 260 of the Discussion have been revised to specifically mention both the interpretation of “negative” results and the potential impact of RGCS on the uptake of other screening tests offered in pregnancy.
3.4 Another point is the legal situation. How are the regulations of genetics tests? Conservative governments might restrict testing, although subgroups of their population might require testing.
Under the Future Research heading the first paragraph has been revised and now refers to the politico-legal and cultural contexts of a region and the potential for government mandates on genetic testing and health care access to influence the offer of RGCS at scale.
Reviewer 3 Report
I really liked this review by the rigor of the selection of material, the structure of the presentation. This review is extremely timely as medical genetics has entered the era of NGS technologies. The main problem is perfectly stated in the review - an insufficient number of specialists who are able to correctly interpret and competently advise on the results.
I have no significant comments. However, it would be great if the authors added a comparison of counseling when using advanced panels and screening for a single disease.
Author Response
I really liked this review by the rigor of the selection of material, the structure of the presentation. This review is extremely timely as medical genetics has entered the era of NGS technologies. The main problem is perfectly stated in the review - an insufficient number of specialists who are able to correctly interpret and competently advise on the results.
I have no significant comments. However,
4.1 it would be great if the authors added a comparison of counseling when using advanced panels and screening for a single disease.
This comparison is now included in the Discussion at line 245.
Round 2
Reviewer 1 Report
Paper is fine now.
Author Response
Thank you for your time and consideration.